# Preclinical In Vitro Studies with 3D Spheroids to Evaluate Cu(DDC)_2_ Containing Liposomes for the Treatment of Neuroblastoma

**DOI:** 10.3390/pharmaceutics13060894

**Published:** 2021-06-17

**Authors:** Friederike Hartwig, Monika Köll-Weber, Regine Süss

**Affiliations:** Department of Pharmaceutical Technology and Biopharmacy, Institute of Pharmaceutical Sciences, University of Freiburg, Sonnenstr. 5, 79104 Freiburg, Germany; monika.koell-weber@pharmazie.uni-freiburg.de (M.K.-W.); regine.suess@pharmazie.uni-freiburg.de (R.S.)

**Keywords:** Disulfiram, Cu(DDC)_2_, liposomes, drug delivery, storage stability, neuroblastoma, 3D spheroids

## Abstract

Preclinical in vitro studies of drug candidates for anticancer therapy are generally conducted on well-established 2D cell models. Unfortunately, these models are unable to mimic the properties of in vivo tumors. However, in vitro 3D models (spheroids) have been proven to be superior in reflecting the tumor microenvironment. Diethyldithiocarbamate (DDC^−^) is the active metabolite of Disulfiram, an approved drug for alcoholism and repurposed for cancer treatment. DDC^−^ binds copper in a molar ratio of 2:1 resulting in a water-insoluble Cu(DDC)_2_ complex exhibiting anticancer activities. Delivery of the Cu(DDC)_2_ complex using nanoparticulate carriers provides decisive advantages for a parental application. In this study, an injectable liposomal Cu(DDC)_2_ formulation was developed and the toxicity was compared with a 2D neuroblastoma and a 3D neuroblastoma cell model. Our results indicate that Cu(DDC)_2_ liposomes complied with the size requirements of nanoparticles for intravenous injection and demonstrated high drug to lipid ratios as well as colloidal stability upon storage. Furthermore, an efficient cytotoxic effect on neuroblastoma 2D cell cultures and a very promising and even more pronounced effect on 3D cell cultures in terms of neuroblastoma monoculture and neuroblastoma co-culture with primary cell lines was proven, highly encouraging the use of Cu(DDC)_2_ liposomes for anticancer therapy.

## 1. Introduction

The development of drug candidates for anticancer therapy needs proper preclinical in vitro and in vivo studies to prevent clinical trial failure. The assessment of drug potency and selectiveness is of particular importance. The drug candidate should be able to reach the target tissue and eliminate cancer cells without significantly affecting non-malignant cells [1]. Regarding in vitro assays, 2D cell monoculture models are most frequently used. These models are easily established, generate reproducible results, and are performable at low costs. They can be applied as a primary testing model using cancer cells, as well as non-malignant cells to assess efficiency, toxicity, and selectiveness of anticancer drugs. However, 2D cell monoculture models are unable to reflect the properties of in vivo tumors and their resistance to drugs [2]. Performing in vitro tests with 2D models unfortunately often leads to false positive testing results and, when being further examined in preclinical steps, to ineffective treatment of tumor xenografts. As an alternative, in vitro multicellular 3D models more accurately represent the microenvironment of in vivo tumors, e.g., allowing cells to secrete extracellular matrix (ECM) components. Cells can interact with the ECM and among each other, affecting cell proliferation, differentiation, gene and protein expression, and resistance to anticancer drugs [3]. 3D cultures can be categorized as scaffold-free or scaffold-based culture systems, where scaffolds can be produced with natural or synthetic material. Scaffold-free systems are often referred to as spheroids [4]. Similar to solid tumors, spheroids display different cell layers building gradients of nutrients, oxygen, and waste. The outer layer consists of highly proliferating cells, the middle layer of quiescent cells and the core, depending on the spheroid size, of necrotic cells [5]. The cellular arrangement and the diffusional limit to mass transport form barriers for drugs and nanoparticles, e.g., liposomes [6,7]. Size, charge, and surface characteristics of nanoparticles can influence electrostatic interactions with the ECM and therefore penetration and uptake into the spheroid [8].

This study combines the evaluation of a novel anticancer drug delivery system and its effectivity in 2D and 3D cell models. Diethyldithiocarbamate (DDC^−^) is described to be the active metabolite of Disulfiram (DSF) and is able to chelate Cu^2+^ in a 2:1 molar ratio resulting in a water insoluble Cu(DDC)_2_ complex, which exhibits anticancer activity [8,9]. DSF, which was approved in 1951 for anti-alcoholism medication, came into focus for anticancer studies where it was shown, that the anticancer activity of DSF, which is rapidly reduced to its active metabolite DDC^−^, strongly depends on the presence of Cu^2+^ [10]. Anyhow, co-administration of DSF and Cu^2+^ results in very low in vivo concentrations of Cu(DDC)_2_, due to poor in vivo stability and rapid degradation of DSF in the presence of serum [8]. The administration of preformed Cu(DDC)_2_ is expected to enhance anticancer effects. However, the poor aqueous solubility of Cu(DDC)_2_, and consequently the insufficient bioavailability results in only low efficiency in preclinical models. Wehbe et al. (2016) published a method to circumvent solubility problems of Cu(DDC)_2_ by using liposomes as nanoscale reaction vessels [8]. Liposomes containing Cu^2+^ in the aqueous core are loaded with DDC^−^, which is able to diffuse through the lipid bilayer. Inside the aqueous core, DDC^−^ chelates Cu^2+^ and precipitates as Cu(DDC)_2_. In this study, the liposome preparation method described by Wehbe et al. (2016) was used with modified purification steps and the introduction of surface area filtration to remove non-encapsulated Cu(DDC)_2_ from Cu(DDC)_2_ liposomes [8]. Non-PEGylated and PEGylated Cu(DDC)_2_ liposomes with a liposomal composition of DSPC:Chol [55:45 molar ratio] and DSPC:Chol:DSPE-mPEG_2000_ [50:45:5 molar ratio] were analyzed for colloidal stability, drug release during storage and cytotoxicity in 2D and 3D cell models. For the latter, an easy-handling and low-cost method was used to generate reproducible spheroids as a sophisticated primary in vitro testing model for anticancer drugs. Therefore, spheroids were cultivated as 3D cancer cell monocultures or 3D co-cultures with primary human cells. Comparative cell experiments were performed with cancer and primary human cell 2D monocultures. Neuroblastoma cells (LS) [11] were used as cancer cell line, which appeared to be highly suitable for spheroid formation.

Neuroblastoma is the most common pediatric extra-cranial solid tumor and accounts for approximately 15% of all cancer-related deaths in infants and children [12,13,14]. Acquiring multidrug resistance is a major obstacle to the successful treatment of neuroblastoma [15,16]. In addition to the administration of multi-chemotherapy regimens to overcome drug resistance, alternative approaches have been explored. Inhibiting the potency of cancer stem cells (CSCs), which are key drivers of tumor progression, metastasis, and drug resistance, is of particular importance [15,17]. It was shown, that aldehyde dehydrogenases (ALDH) are important key players to maintain the role of CSCs [17], which includes inhibitors of ALDH as potential candidates for new therapeutic approaches. Recent studies show effective inhibition of ALDH through DSF as well as Cu(DDC)_2_ treatment [18,19,20,21]. In this context, Cu(DDC)_2_ is highlighted as a potential candidate. The use of a liposomal and parenterally injectable Cu(DDC)_2_-formulation is expected to be an alternative approach for the treatment of neuroblastoma in comparison to multi-chemotherapy regimens. Cytotoxic effects of a liposomal Cu(DDC)_2_-formulation are evaluated with in vitro neuroblastoma 3D spheroids. This 3D in vitro system suggests a more appropriate cell model that significantly better simulates the physiological conditions of a tumor as compared to a 2D cell model.

## 2. Materials and Methods

### 2.1. Materials

1,2-distearoyl-sn-glycero-3-phosphocholine (DSPC) and 1,2-distearoyl-sn-glycero-3-phosphoethanolamine-N-[methoxy(polyethylene glycerol)-2000] (DSPE-mPEG_2000_) were generously donated by Lipoid GmbH (Ludwigshafen, Germany). Cholesterol and 3,3′-dioctadecyloxacarbocyanin perchlorate (DiO, catalog number 42364) were purchased from Sigma Aldrich (Darmstadt, Germany). RPMI 1640 with (cat. no. P04-18500) and without (catalog number P04-16520) phenol red, DMEM (catalog number P04-04500), sodium pyruvate 10 nM, PBS w/o Ca^2+^ and Mg^2+^, Trypsin/EDTA 0.05%/0.02% and 0.25%/0.02% were obtained from PAN Biotech (Aidenbach, Germany). Fetal calf serum (FCS) was purchased from Sigma Aldrich. Sephadex G-50 was obtained from GE Healthcare Life Science (Marlborough, MA, USA), CellTiter Glo^®^ 2D and 3D assays (catalog number G7572 and G9683) from Promega (Walldorf, Germany), calcein-AM (catalog number C3099) from Molecular Probes (Eugene, OR, USA), and propidium iodide (PI, catalog number CN74.1) from Carl Roth (Karlsruhe, Germany). Copper sulphate, HEPES, sodium diethyldithiocarbamate trihydrate, sucrose, and all other chemicals were purchased from Carl Roth (Germany). Vivaspin Turbo 4, filtration unit, 100,000 MWCO (catalog number VSO4T42) was purchased from Sartorius and sterile cellulose acetate filter, 0.2 µm and 0.45 µm from VWR (International, Radnor, PA, USA).

### 2.2. Liposomal Preparation

The preparation of Cu(DDC)_2_ liposomes followed the idea of Wehbe et al. (2016) and was modified as indicated (Figure 1) [8]. Liposomes composed of DSPC:Chol [55:45 molar ratio] and DSPC:Chol:DSPE-mPEG_2000_ [50:45:5 molar ratio] were prepared by the thin film hydration method [22] with subsequent hand extrusion [23]. Briefly, organic stock solutions were transferred at the indicated ratios into a round bottom flask and the solvent was removed via rotary evaporation. The flask was placed under vacuum for at least 2 h to remove residual solvent. The lipid film was hydrated with 1 mL of an aqueous CuSO_4_ solution (Cu^2+^, 150 mM in aqua destillata, pH 3.5) resulting in a lipid concentration of 40 mM. After 30 min of swelling and agitation, the formed liposomes were extruded for 41 passages through a polycarbonate membrane with a pore diameter of 80 nm (Nuclepore^®^, GE Healthcare Life Science, Chicago, IL, USA). Solvent removal, film hydration, and extrusion were performed at 65 °C which is 10 °C above the phase transition temperature of DSPC. Prior to the addition of DDC^−^, Cu^2+^ liposomes were separated from non-encapsulated Cu^2+^ via size exclusion chromatography (SEC). The SEC column, prepared with Sephadex G-50, was equilibrated with an EDTA containing sucrose buffer (SHE, 300 mM sucrose, 20 mM HEPES, 30 mM EDTA, pH 7.4). Hereafter, buffer exchange with an EDTA-free sucrose buffer (SH, 300 mM sucrose, 20 mM HEPES, pH 7.4) was performed via three centrifugation steps at room temperature (RT) and 3000× *g* for 1 h using Vivaspin^®^ Turbo 4 filtration units (100 kDa MWCO, Sartorius, Germany). To allow the diffusion of DDC^−^ into the liposomal interior, followed by complexation of DDC^−^ with the liposomal encapsulated Cu^2+^, the preparation was left at 25 °C for 10 min at 300 rpm in a shaker. Free DDC^−^ was removed via filtration units as described above with the final SH formulation buffer. The resulting Cu(DDC)_2_ liposomes were prefiltered via a cellulose acetate (CA) 0.45 µm filter (VWR International, USA) to remove non-incorporated and precipitated Cu(DDC)_2_. For cell experiments, the liposomes were sterile filtered under aseptic conditions through a 0.2-µm CA filter. Fluorescent-labeled liposomes (with 0.5% DiO, related to the total lipid amount) were used to perform flow cytometry experiments. For the DiO liposome preparation, DiO was added from an organic stock solution to the residual liposomal components in a round bottom flask. The lipid film was treated as described above. In this case, the lipid film was hydrated with 1 mL HEPES buffered saline (HBS, 10 mM HEPES, 140 mM NaCl, pH 7.4), resulting in a final lipid concentration of 20 mM. The liposomal dispersion was extruded through an 80-nm polycarbonate membrane (Nuclepore^®^, GE Healthcare Life Science, USA) after 30 min of swelling and agitation and stored at 4–6 °C protected from light until use.

### 2.3. Surface Area Filtration

The usage of surface area filtration had to be validated to remove non-incorporated and precipitated Cu(DDC)_2_ from liposomal Cu(DDC)_2_. For this purpose, freshly prepared Cu(DDC)_2_ liposomes were prefiltered through a 0.45-µm CA filter and sterile-filtered through a 0.2-µm CA filter. These filtration steps are part of the liposomal production process and are referred to as the 1st + 2nd filtrations. To validate the influence of filtration steps on liposomal stability, two additional filtration steps were performed referred to as the 3rd and 4th filtrations. Filtered samples were taken after each filtration step for analyzing lipid and Cu(DDC)_2_ content.

### 2.4. Quantification of Lipids and Cu(DDC)_2_

The Cu(DDC)_2_ liposomes were analyzed for total lipid and Cu(DDC)_2_ content. Phospholipid content was determined via Bartlett assay [24] and Cu(DDC)_2_ content via a spectrophotometric method. Quantification of DDC^−^ was determined by measuring the absorbance of complexed Cu^2+^ at a wavelength of λ_max_ = 435 nm with a GENESYS 10S UV-Vis spectrophotometer (Thermo Scientific™, Waltham, MA, USA). Standards were prepared by mixing Cu^2+^ solutions (0.015–0.1 µmol Cu^2+^ in aqua dest., pH 3.5) with a 70-µmol methanolic DDC^−^ stock solution resulting in the formation of solubilized Cu(DDC)_2_. The calibration curve was obtained by plotting the absorbance versus the amount of substance of Cu^2+^. DDC^−^ chelates Cu^2+^ in a 2:1 molar ratio and, therefore, the amount of complexed DDC^−^ can be calculated. For the analysis of Cu(DDC)_2_ liposomes, the use of methanol has the advantage of disrupting the liposomal membrane and solubilizing the liposomal trapped Cu(DDC)_2_. Subsequently, the absorbance of complex Cu^2+^ and accordingly the amount of Cu(DDC)_2_ can be determined via UV-Vis spectrometry. Linearity was verified by linear least square regression analysis, where *y* is the absorbance, *x* the amount of substance of sample, *m* the slope, and *b* the y intercept:(1)y=mx+b

In this study, the Cu(DDC)_2_ concentrations of Cu(DDC)_2_ liposomes refer to the amount of intraliposomal complexed DDC^−^, calculated as described above. Accordingly, the drug concentration should be indicated with DDC^−^/Cu^2+^, whereby the molar ratio of DDC^−^ + 0.5 mol Cu^2+^ should be noted. For simplicity, DDC^−^/Cu^2+^ liposomes were entitled as Cu(DDC)_2_ liposomes. This is justified by the fact, that Cu(DDC)_2_ is formed by DDC^−^ and Cu^2+^ in the aqueous core of liposomes and that Cu(DDC)_2_ is responsible for the anticancer activity.

### 2.5. Liposome Characterization and Stability Measurement

#### 2.5.1. Size and Polydispersity Index

The size expressed as the hydrodynamic diameter (d_h_) and the polydispersity index (PDI) were measured via dynamic light scattering (DLS, ZetaPals, Brookhaven Instruments Corporation, Holtsville, NY, USA). Liposomal samples were diluted to 1–5 mM (total lipid) in filtered SH buffer for size and polydispersity analyses.

#### 2.5.2. Cu(DDC)_2_ to Lipid Ratio

The Cu(DDC)_2_ to lipid ratio (drug to lipid ratio, *D/L*) was calculated with Equation (2), where *n_entrapped drug_* is the amount of substance of complexed DDC^−^ with Cu^2+^ and *n_total lipid_* the amount of substance of lipid: (2)$$D/L\;\left[\mathrm{mol}:\mathrm{mol}\right]=n_{entrapped\;drug}/n_{total\;lipid}$$

#### 2.5.3. Liposomal Stability Studies

For storage stability studies the liposomes were kept at 4–6 °C for 6 months (183 days) and at RT (22 ± 2 °C) and −20 °C for 2 months (61 days). Therefore, aliquots of non-PEGylated and PEGylated Cu(DDC)_2_ liposomes were prepared and characterized via DLS at the indicated time points. Stability studies included the determination of the release rate of Cu(DDC)_2_ from liposomes after storage of samples at 4–6 °C for 6 months using surface area filtration (Section 2.3). 

### 2.6. Cell Culture

LS, Kelly, and SH-SY5Y neuroblastoma cells, as well as primary human skeletal muscle cells (hSkMC) and primary human dermal fibroblasts, and adult (hDFa) were obtained as a kind gift from Professor Dr. Rupert Handgretinger (Universitätskinderklinik Tübingen, Germany). Cells were cultivated at 37 °C, 5% CO_2_ in a humidified incubator in RPMI supplemented with 10% FCS (LS, Kelly, SH-SY5Y, and hDFa) or DMEM supplemented with 5% sodium pyruvate and 10% FCS (hSkMC).

Next, 3D cell cultures were generated by the liquid overlay method [25]. Briefly, each well of a 96-well microtiter plate was coated under aseptic conditions with 50 µL of an autoclaved and therefore liquefied 1.5% (*w*/*v*) agarose–PBS solution. After cool down (to RT) and solidification of the agarose, 200 µL of a 20,000 cell/mL suspension was added per well. Following a centrifugation step at 600× *g*, for 5 min at 20 °C, the plate was incubated for 72 h in an incubator (37 °C and 5% CO_2_, humidified) until reaching a spheroid diameter of approximately 400 µm to start the liposomal treatment. Spheroid formation and growth were documented with an Axiovert 40 CFL microscope, an Axiocam 305 color camera, and a ZEN core v3.0 software via extended depth of focus (Carl Zeiss, Oberkochen, Germany). For co-culture spheroids, the cell number of a primary and a cancer cell dispersion was adjusted to a mixed cellular dispersion at a 1:1 ratio. LS-hDFa co-culture spheroids were cultivated in RPMI supplemented with 10% FCS and LS-hSkMC co-culture spheroids were cultivated in DMEM supplemented with 5% sodium pyruvate and 10% FCS.

### 2.7. Cell Viability Experiments

For 2D monolayer experiments neuroblastoma cells and primary cells were seeded in white opaque 96-well microtiter plates (Greiner Bio, Frickenhausen, Germany) in a density of 2000–7000 cells per well and cultured overnight. Cells were treated with Cu(DDC)_2_ liposomes, free Cu(DDC)_2_ and Cu^2+^. After 24, 48, or 72 h of treatment, CellTiter-Glo^®^ for 2D cell cultures was added according to the manufacturer’s protocol. For 3D spheroid experiments, 3D cell cultures were generated as described in Section 2.6. Following 3 days of incubation in an incubator, spheroids were treated with Cu(DDC)_2_ liposomes. After 72 h, the viability was determined with the CellTiter-Glo^®^ assay for 3D cell cultures. EC_50_ (half maximal effective concentration) values were calculated by Excel solver and non-linear least squares fitting [26] with the following equation.
(3)$$y=y_{0\;}+\frac{y_{100}-y_{0}}{1+{\left(\frac{EC_{50}}{x}\right)}^{-p}}$$

The cell viability (*y*) is calculated as concentration (*x*) in µM. The term *y*_100_ represents the viability of control cells, *y*_0_ the lowest viability, and *p* the hill coefficient [27]. Cytotoxicity analyses were evaluated with at least three different data sets. The *EC*_50_ value is calculated as the mean of the generated *EC*_50_ values.

### 2.8. Epifluorescent Staining of 3D Spheroids

LS spheroids were cultured in RPMI 1640 medium w/o phenol red and treated with non-PEGylated or PEGylated Cu(DDC)_2_ liposomes. Spheroids were stained with 1 µM calcein-AM and 5 µM PI for the last 2 h of a 72 h incubation period of Cu(DDC)_2_ liposomes. Living and dead cells were analyzed with an Axiocam 305 color camera, a ZEN core v3.0 software (Carl Zeiss, Oberkochen, Germany), a 10× objective and a corresponding filter set for calcein-AM and propidium iodide.

### 2.9. Cellular Uptake Studies

Cellular association and uptake of non-PEGylated and PEGylated DiO liposomes were assessed via flow cytometry (BD LSRFortessa™, Becton Dickinson, Kelberg, Germany). DSPC:Chol:DiO [55:45 + 0.5 molar ratio] and DSPC:Chol:DSPE-mPEG_2000_:DiO [50:45:5 + 0.5 molar ratio] liposomes, carrying DiO as the fluorescent label, were diluted with culture medium to reach a final lipid concentration of 0.75 mM. LS cells were seeded in 24 well plates at a density of 60,000 cells/mL and cultured overnight. Cells were incubated for 15 min, 1, 2, 4, and 6 h with DiO-labelled liposomes. After the incubation periods, cells were washed with ice cold PBS w/o Ca^2+^ and Mg^2+^, trypsinized and detached from wells via resuspension in PBS buffer and collected in FACS tubes. The cells were analyzed using a 488-nm blue laser for excitation and a 530/30-nm filter set. There were 10,000 events per sample recorded and cells were gated using forward and sideward scattering to determine the live cell population. Afterwards, the extracellular DiO fluorescence was quenched by the addition of 0.08% trypan blue [28] and the samples were repeatedly analyzed to distinguish cellular association and cellular uptake of DiO-labelled liposomes. The fluorescence intensity and the percentage of fluorescent cells (living gate) were evaluated.

### 2.10. Statistical Analysis

All data are expressed as mean ± SD. GraphPad Prism 8.01 software was used for statistical analysis. One-way ANOVA with Tukey’s multiple comparison test was used to evaluate differences between unpaired data sets. One-way ANOVA with Dunnett’s multiple comparison test was used to evaluate differences between the mean of each sample with the mean of the control (unpaired). Differences were considered significant when *p* < 0.05 (* < 0.05, ** < 0.01, *** < 0.001) in all analyses.

## 3. Results

### 3.1. Stability of Cu(DDC)_2_ Liposomes

#### 3.1.1. Validation of Non-Encapsulated Cu(DDC)_2_ Precipitate Removal via Surface Area Filtration

The preparation process of Cu(DDC)_2_ liposomes and the experimental set up for stability analyses was validated. Initially, liposomal agglomeration occurring after the preparation process of Cu(DDC)_2_ liposomes was noticed via DLS and Cryo-TEM (data not shown). This was mainly due to the presence of extraliposomal (non-encapsulated) precipitated Cu(DDC)_2_ which agglomerates with liposomes due to the lipophilicity of the Cu(DDC)_2_ complex. Therefore, surface area filtration was introduced as the final step in the preparation process to retain extraliposomal Cu(DDC)_2_ from Cu(DDC)_2_ liposomes. This approach had to be validated to ensure extraliposomal Cu(DDC)_2_ retention without lipid loss. Figure 2a shows a schematic representation of the validation process with 1^st^ + 2^nd^ filtration as part of the liposomal preparation process and a 3^rd^ and 4^th^ filtration to test consecutive filtration without affecting the D/L ratio and reducing the lipid amount of the liposomal dispersion. The results were evaluated statistically via one-way ANOVA and Tukey’s multiple comparison test. The validation analysis was performed with non-PEGylated (DSPC:Chol [55:45 molar ratio]) and PEGylated (DSPC:Chol:DSPE-mPEG_2000_ [50:45:5 molar ratio]) Cu(DDC)_2_ liposomes (Figure 2b). A reduction of the D/L ratio was noted before and after 1^st^ + 2^nd^ filtration for non-PEGylated and PEGylated Cu(DDC)_2_ liposomes, whereas the lipid amount remained constant. No significant differences in D/L ratios and lipid amounts were detected for both formulations between the 1^st^ + 2^nd^ filtration and 3^rd^ filtration. Non-PEGylated liposomes revealed a slight significant difference among the D/L ratios (*p* = 0.0104) between the 1^st^ + 2^nd^ filtration and 4^th^ filtration. Overall, the lipid amount did not change for neither non-PEGylated nor PEGylated Cu(DDC)_2_ liposomes during consecutive filtration through 0.2-µm CA filters and no significant change of the D/L ratio was observed after the 3^rd^ filtration step. 

#### 3.1.2. In Vitro Colloidal Stability and Drug Release Analysis of Cu(DDC)_2_ Liposomes

As described in Section 2.2, non-PEGylated and PEGylated Cu(DDC)_2_ liposomes were prepared via the thin film hydration method with subsequent extrusion. Non-PEGylated Cu(DDC)_2_ liposomes showed a d_h_ of 156 ± 7 nm, a PDI of 0.16 ± 0.02, and a D/L ratio of 0.15 ± 0.03 mol:mol. PEGylated Cu(DDC)_2_ liposomes exhibited a d_h_ of 161 ± 7 nm, a PDI of 0.14 ± 0.01, and a D/L ratio of 0.30 ± 0.04 mol:mol (Table 1).

Particle size, size distribution, and drug retention are critical parameters during storage. Therefore, aliquots of Cu(DDC)_2_ liposomes were stored (in SH buffer) for 6 months (183 days) at 4–6 °C (Figure 3a) and for 2 months (61 days) at RT (22 ± 2 °C; Figure 3b) and −20 °C (Figure 3c) and analyzed for size and size distribution. Data was evaluated statistically using one-way ANOVA and Dunnett’s post-hoc test by comparing the mean of each d_h_-column with the mean of the control d_h_-column (t = 0 days). For Cu(DDC)_2_ liposomes, stored at 4–6 °C, a significant difference in d_h_ was observed after 3 months (91 days) for non-PEGylated (*p* < 0.001) and after 6 months for PEGylated Cu(DDC)_2_ liposomes (*p* = 0.006) compared to freshly prepared preparations. Figure 3b shows initial instabilities after 14 days for non-PEGylated (*p* = 0.02) and after 28 days for PEGylated Cu(DDC)_2_ liposomes (*p* = 0.01) stored at RT. After one day storage at −20 °C non-PEGylated (*p* < 0.001) and PEGylated (*p* < 0.001) Cu(DDC)_2_ liposomes exhibited increasing instability. The storage temperature of 4–6 °C resulted in constant size and size distribution for non-PEGylated and PEGylated Cu(DDC)_2_ liposomes.

After the determination of a suitable storage temperature, it had to be proven that Cu(DDC)_2_ does not leak out of liposomes in the timeframe between preparation, storage, and actual administration. Therefore, aliquots of non-PEGylated and PEGylated Cu(DDC)_2_ liposomes were adjusted to equal D/L ratios and stored for 6 months at 4–6 °C. The liposomal aliquots were refiltrated through a 0.2-µm CA filter after specific storage periods and the D/L ratio of the filtrates was determined. A decrease in D/L ratio indicated Cu(DDC)_2_ release. The mean D/L ratio of t = 0 days was defined as control (100%), so the collected data is expressed as % of control and plotted against the time (t = 0–183 days; Figure 4). PEGylated Cu(DDC)_2_ liposomes displayed 90–100% drug retention until 3 months of storage, which decreased to 74% after 6 months of storage. For non-PEGylated Cu(DDC)_2_ liposomes the drug retention decreased to 54% after 7 days and furthermore to 15% after 6 months.

### 3.2. In Vitro Determination of the Cytotoxicity of Cu(DDC)_2_ Liposomes, Free Cu^2+^ and Cu(DDC)_2_, and the Evaluation of Liposomal Uptake in Neuroblastoma 2D Monolayers 

The cytotoxic potency of Cu(DDC)_2_ liposomes was compared to free Cu(DDC)_2_ and free Cu^2+^. Therefore, cell viability was determined in Kelly and SH-SY5Y (neuroblastoma cells) using the 2D CellTiter-Glo^®^ assay (Figure 5a,b). EC_50_ values of liposomal Cu(DDC)_2_, free Cu(DDC)_2_, and Cu^2+^ achieved with Kelly cells were in a concentration range of 0.12 ± 0.01 µM, 0.42 ± 0.03 µM, and 427.19 ± 8.86 µM and for SH-SY5Y in a concentration range of 0.13 ± 0.02 µM, 0.37 ± 0.09 µM, and 483.80 ± 62.07 µM, respectively. Notably, the cytotoxic effect of liposomal Cu(DDC)_2_ was higher than free Cu(DDC)_2_ on neuroblastoma cell lines (EC_50_ of Cu(DDC)_2_ liposomes: 0.12 ± 0.01 µM (Kelly), 0.13 ± 0.02 µM (SH-SY5Y); EC_50_ of free Cu(DDC)_2_: 0.42 ± 0.03 µM (Kelly), and 0.37 ± 0.09 µM (SH-SY5Y)). Cu^2+^ revealed cytotoxic effects in a concentration range significantly exceeding the applied Cu^2+^ concentration in liposomal Cu(DDC)_2_ formulations (Figure 5c,d).

Furthermore, in vitro cellular uptake experiments were conducted on LS monolayers with 0.75 mM non-PEGylated and PEGylated DiO liposomes and determined via flow cytometry (Figure 6). Cells were treated for 15 min, 1, 2, 4, and 6 h with DiO liposomes. The cellular uptake was rather low in the first hours and reached 24.8 ± 5.2% for non-PEGylated DiO liposomes and 17.8 ± 3.0% for PEGylated DiO liposomes after 6 h of incubation. A similar cellular fluorescence intensity was measured for both formulations.

### 3.3. Cytotoxicity and Selectivity of Cu(DDC)_2_ Liposomes on 2D and 3D Cell Cultures

Drug potency and selectivity of Cu(DDC)_2_ liposomes were evaluated with LS neuroblastoma and primary hSkMC and hDFa cells using 2D monolayers and 3D spheroids. Initially, the cytotoxic potency was analyzed in 2D LS monolayers after 24, 48, and 72 h of incubation with non-PEGylated and PEGylated Cu(DDC)_2_ liposomes (Figure 7). The cell viability was quantified via the 2D CellTiter-Glo^®^ assay. Statistical analyzes were performed with one-way ANOVA and Tukey’s multiple comparison test. The resulting EC_50_ values for 24, 48, and 72 h were in a concentration range of 0.16 ± 0.01 µM, 0.06 ± 0.01 µM, and 0.05 ± 0.02 µM for non-PEGylated and 0.13 ± 0.01 µM, 0.04 µM, and 0.07 ± 0.02 µM for PEGylated Cu(DDC)_2_ liposomes, respectively. Cu(DDC)_2_ liposomes affected the cell viability in a concentration and time dependent manner (Figure 7a,b), whereby the calculated EC_50_ values after 48 and 72 h of treatment did not vary significantly (48 vs. 72 h EC_50_ of non-PEGlyated liposomes: *p* = 0.71; 48 vs. 72 h EC_50_ of PEGylated liposomes: *p* = 0.21; Figure 7c,d). No difference in cytotoxic potency was observed comparing non-PEGylated and PEGylated Cu(DDC)_2_ liposomes.

Furthermore, cell viability of 3D LS spheroids was analyzed 72 h after treatment with Cu(DDC)_2_ liposomes via the 3D CellTiter-Glo^®^ assay. 3D spheroids were generated as described in Section 2.6 via the liquid overlay technique resulting in dense spheroids with a highly reproducible diameter and an inter plate coefficient of variation less than 3% [27]. Cu(DDC)_2_ liposomes affected the cell viability of 3D LS spheroids dose-dependently (Figure 8a,b). Non-PEGylated Cu(DDC)_2_ liposomes revealed an EC_50_ value of 4.09 ± 0.01 µM and PEGylated Cu(DDC)_2_ liposomes of 3.95 ± 0.80 µM when used on 3D LS spheroids. Hence, the resulting EC_50_ values were far higher in comparison to those of 2D LS monolayers (Figure 8c,d). Microscopy images of 3D LS spheroids were taken over time during treatment with Cu(DDC)_2_ liposomes (Figure 8e,f). Surprisingly, treatment with non-PEGylated liposomes led to complete disintegration of the spheroids, whereby treatment with PEGylated liposomes resulted in maintenance of the spherical shape.

Live/dead staining of 3D LS spheroids allowed to distinguish between proliferating and dead cells to further evaluate the influence of Cu(DDC)_2_ liposomes (Figure 9). Therefore, 3D LS spheroids were stained with calcein-AM and PI for the last 2 h of a 72 h incubation period with Cu(DDC)_2_ liposomes and compared with untreated spheroids being incubated with cell medium for the same duration of time (72 h total). The untreated control revealed a typical architecture of a dense spheroid containing a necrotic core (red) surrounded by an outer rim consisting of proliferating cells (green; Figure 9). This cellular arrangement changed dose dependently after treatment with Cu(DDC)_2_ liposomes, resulting in an increasing portion of dead cells, according to the viability analyses (Figure 8). Single detached dead cells were visible after administration of low liposomal Cu(DDC)_2_ dosages. With increasing concentration of Cu(DDC)_2_ liposomes the PI signal intensified in the outer proliferating cell layer until it reached the necrotic core (Figure 9).

In a next step, the cytotoxicities of non-PEGylated and PEGylated Cu(DDC)_2_ liposomes were examined by using non-malignant cells. 2D monolayers of hSkMC as well as hDFa were incubated for 24 h with Cu(DDC)_2_ liposomes. The cell viability was analyzed via the 2D CellTiter-Glo^®^ assay (Figure 10a,b) and compared to the cell viability of treated LS monolayers. Calculated EC_50_ values (Figure 10c,d) were analyzed statistically via one-way ANOVA and Tukey’s multiple comparison test. hSkMC and hDFa displayed significantly higher EC_50_ values in comparison to LS cells after treatment with Cu(DDC)_2_ liposomes (hSkMC vs. LS with non-PEGylated liposomes: *p* < 0.001 and PEGylated liposomes: *p* = 0.004; hDFa vs. LS with non-PEGylated liposomes: *p* < 0.001 and PEGylated liposomes: *p* < 0.001). The hDFa cells displayed higher EC_50_ values compared to hSkMC cells (EC_50_ of non-PEGylated Cu(DDC)_2_ liposomes: 0.28 ± 0.01 µM (hSkMC) and 0.51 ± 0.03 µM (hDFa); EC_50_ of PEGylated Cu(DDC)_2_ liposomes: 0.38 ± 0.07 µM (hSkMC) and 0.61 ± 0.02 µM (hDFa); Figure 10c,d). Moreover, hSkMC and hDFa were more sensitive to non-PEGylated compared to PEGylated Cu(DDC)_2_ liposomes. Anyhow, the viability of non-malignant cells was affected by Cu(DDC)_2_ liposomes in a dose dependent manner.

3D LS spheroids were also generated as co-culture with hSkMC as well as hDFa to analyze the cytotoxic effectiveness of Cu(DDC)_2_ liposomes. Consequently, 3D co-culture spheroids were also treated for 72 h with non-PEGylated and PEGylated Cu(DDC)_2_ liposomes. The viability was determined via the 3D CellTiter Glo^®^ assay, resulting in similar cytotoxic responses as determined in the tested cell lines (Figure 11a,b). Statistical analysis between EC_50_ values via one-way ANOVA and Tukey’s multiple comparison test confirmed the absence of significance (EC_50_ values from LS vs. LS-hSkMC vs. LS-hDFa: *p* = 0.46 (non-PEGylated Cu(DDC)_2_ liposomes), *p* = 0.48 (PEGylated Cu(DDC)_2_ liposomes); Figure 11c,d). Figure 11e,f depicts brightfield images of mono- and co-culture spheroids after 72 h of treatment with Cu(DDC)_2_ liposomes. According to Figure 8e,f, the same morphological changes of 3D spheroids can be observed after treatment with Cu(DDC)_2_ liposomes. Treatment with non-PEGylated Cu(DDC)_2_ liposomes led to spheroid disintegration, whereby treatment with PEGylated Cu(DDC)_2_ liposomes resulted in the maintenance of the spherical shape.

## 4. Discussion and Outlook

In this study, liposomes were used as a drug delivery system for the water-insoluble Cu(DDC)_2_ complex which exhibits cytotoxic effects on cancer cells. A reproducible preparation process of Cu(DDC)_2_ liposomes was developed on the basis of the publication of Wehbe et al. (2016) [8]. The first part of this paper describes the additional integration of surface area filtration into the liposomal preparation process as a suitable method to remove extraliposomal Cu(DDC)_2_ precipitates from Cu(DDC)_2_ liposomes without lipid loss. It was proven that further consecutive filtration steps had no impact on the D/L ratio of Cu(DDC)_2_ liposomes. Therefore, surface area filtration might also be considered as an in-process control step to evaluate and retain eventually released Cu(DDC)_2_ from Cu(DDC)_2_ liposomes after stress exposure (e.g., heat or storage), without affecting the D/L ratio by itself. Furthermore, the colloidal stability and drug release upon storage of non-PEGylated (DSPC:Chol [55:45 molar ratio]) and PEGylated (DSPC:Chol:DSPE-mPEG_2000_ [50:45:5 molar ratio]) Cu(DDC)_2_ liposomes were compared. In general, the physical stability of a liposomal dispersion is affected by changes in liposomal size due to aggregation and fusion and the leakage of entrapped drug. These physical instabilities originate from oxidation and hydrolysis of the liposomal constituents. Hence, the determination of the liposomal size and size distribution is a sensitive indicator of the physical liposomal stability [29]. The colloidal stability of Cu(DDC)_2_ liposomes were evaluated at different storage temperatures (4–6 °C, RT, and −20 °C) in the presence of an SH buffer. An ideal storage temperature of 4–6 °C for liposomes was confirmed in various studies [30,31]. Furthermore, in this study, it was shown that a constant colloidal stability of non-PEGylated and PEGylated Cu(DDC)_2_ liposomes were achieved at a storage temperature of 4–6 °C. Increasing instabilities were observed while storing at RT and −20 °C, despite the presence of a cryoprotective disaccharide (sucrose) in the liposomal buffer. In particular, storage at −20 °C could lead to mechanical stresses due to the formation of ice crystals during freezing. Buffers with glycerol and carbohydrates (e.g., trehalose) are reported to be more suitable for freezing liposomal dispersions in a temperature range between −20 and −30 °C [32]. However, it is recommended to store Cu(DDC)_2_ liposomes at 4–6 °C, since this condition already reduces temperature-dependent hydrolysis and further cooling down of the preparation did not improve stability [29]. Moreover, the integration of hydrophilic PEG polymers on the surface of Cu(DDC)_2_ liposomes not only resulted in higher D/L ratios, it also improved the colloidal stability during storage at 4–6 °C, probable through steric [33] and electrostatic repulsion [34]. Also, PEGylated Cu(DDC)_2_ liposomes displayed a higher drug retention upon storage at 4–6 °C in comparison to non-PEGylated Cu(DDC)_2_ liposomes. Thus, the usage of PEGylated Cu(DDC)_2_ liposomes is recommended due to higher membrane stability. Overall, the presence of DSPE-mPEG_2000_ can significantly modify characteristics of a liposomal formulation (e.g., increased entrapment efficiencies and higher drug retentions) [35]. In future studies, further impacts (e.g., pH, oxygen, and light) have to be examined to fully analyze the stability and degradation profile of Cu(DDC)_2_ liposomes during storage. In addition, it is recommended to support stability analysis (conducted via DLS) with Cryo-TEM images. For long term storage of Cu(DDC)_2_ liposomes, lyophilization can be taken into consideration. Furthermore, serum stability should be tested to further evaluate Cu(DDC)_2_ liposomes for future in vivo experiments. 

In the second part of this study, the in vitro cytotoxicity of Cu(DDC)_2_ liposomes on 2D and 3D neuroblastoma cell models was examined. Initially it was shown, that liposomal Cu(DDC)_2_ was more cytotoxic for 2D neuroblastoma cell cultures than free Cu(DDC)_2_, which highlights the usage of a liposomal Cu(DDC)_2_ formulation. Furthermore, the viability reduction of neuroblastoma cells could be attributed to the Cu(DDC)_2_ complex, since the corresponding concentration of free Cu^2+^ did not affect the cell viability by itself. Beyond that, non-PEGylated and PEGylated Cu(DDC)_2_ liposomes efficiently affected the viability of 2D LS monolayers in a time- and concentration-dependent manner. However, the resulting EC_50_ values achieved for both formulations after 48 and 72 h of liposomal incubation did not differ significantly. Tawari et al. (2015) claimed, that Cu(DDC)_2_ induces the cell death in a direct phase and in a delayed phase [36]. In the direct phase, cell viability is reduced via the formation of reactive oxygen species (ROS) which damages cells [18,34]. In the delayed phase, Cu(DDC)_2_ interferes with essential molecular signaling pathways inducing apoptosis [36]. Consequently, the cellular ATP level rises, since the induction of apoptosis requires energy [37]. Therefore, it is possible, that the cell viability (measured via an ATP-dependent assay) is not significantly reduced after 72 h of incubation compared to 48 h of incubation with Cu(DDC)_2_ liposomes, due to the delayed phase of the cell death induction triggered by Cu(DDC)_2_.

The cytotoxic effectiveness of Cu(DDC)_2_ liposomes were further determined in more complex 3D LS monoculture spheroids. Surprisingly, treatment with non-PEGylated and PEGylated Cu(DDC)_2_ liposomes lead to a distinct change in spheroid morphology. During treatment with non-PEGylated Cu(DDC)_2_ liposomes a disintegration of the spheroid structure was observed where dead cells detached from the spheroid. No disintegration of the spheroid morphology was observed during the treatment of PEGylated Cu(DDC)_2_ liposomes. However, the treatment with non-PEGylated Cu(DDC)_2_ liposomes leads to the same extent of a complete viability reduction throughout different cell layers of the spheroid as with PEGylated Cu(DDC)_2_ liposomes. This outcome implies high drug effectivity of the Cu(DDC)_2_ formulations. Anyhow, a question arises about how the treatment with both Cu(DDC)_2_ formulations can result in different spheroid morphologies. On the one hand, it is reported, that PEGylated liposomes could not interact with the spheroid due to PEG chains hindering an intratumoral transport [38]. This supports the suggestion that the different interactions of non-PEGylated and PEGylated Cu(DDC)_2_ liposomes with cells or the spheroid could lead to distinct spheroid morphologies after treatment. On the other hand, Niora et al. (2020) described that PEGylated liposomes penetrate better into deeper cell layers of a spheroid due to poor interactions with cells compared to non-PEGylated liposomes [39]. Therefore, it is not clear whether non-PEGylated and PEGylated Cu(DDC)_2_ liposomes accumulate at the spheroid surface or whether Cu(DDC)_2_ liposomes are able to diffuse into deeper cell layers. Situation-dependent distinct cell deaths might be induced which could influence cell–cell-contacts and therefore the spheroid morphology. Consequently, differences in the induction of apoptosis and necrosis might be analyzed after spheroid treatment with non-PEGylated and PEGylated Cu(DDC)_2_ liposomes. Particularly apoptosis leads to a dissolution of cellular binding properties [40]. The usage of light sheet fluorescence microscopy could give further information on the penetration behavior of fluorescence-labeled Cu(DDC)_2_ liposomes in 3D spheroids [6,7]. Nonetheless, far higher EC_50_ values were obtained as expected after Cu(DDC)_2_ liposome treatment of 3D LS monoculture spheroids compared to those of 2D LS monolayers. Reasons for the drug resistance of 3D LS spheroids are obviously attributed to specific cellular interactions of the spheroid, to the three-dimensional arrangement of heterogeneous cell populations and the limited diffusion of mass transport which hinders the penetration of drugs and nanoparticles [6,7]. Moreover, cells of a spheroid are able to secrete components of an extracellular matrix (ECM) [3]. Desoize et al. (1998) described how cellular interactions with the microenvironment (cell–cell-contact and cell–ECM-contact) play a crucial role in resistance development against anticancer agents [41]. Presumably, a combination of the before mentioned factors leads to the resistance of 3D LS spheroids against Cu(DDC)_2_ liposomes. Still, Cu(DDC)_2_ liposomes effectively reduced the cell viability throughout different cell layers of the spheroid.

An effective drug candidate for cancer therapy should be able to reach the target tissue and to eliminate cancer cells without affecting non-malignant cells. Therefore, in the third part of this study the in vitro cytotoxicity of Cu(DDC)_2_ liposomes on 2D primary cell models (hSkMC and hDFa) and 3D co-culture cell models (composed of neuroblastoma and primary cells) was analyzed. Here it was shown, that 2D hSkMC and 2D hDFa monolayers were less sensitive towards Cu(DDC)_2_ liposomes compared to 2D LS monolayers. Likewise, Wehbe et al. (2016) observed in vitro a less cytotoxic effect of Cu(DDC)_2_ liposomes on a primary cell line compared to cancer cells [8]. A possible explanation for the outcome could be a cell line-specific internalization rate for liposomes, since the uptake rate of nanoparticles was reported to be lower for non-malignant cells than for cancer cells [42,43]. Thus, the internalization rates of hSkMC and hDFa should be analyzed via uptake experiments similar to Section 3.2. Nevertheless, Cu(DDC)_2_ liposomes turned out to reduce the cell viability of primary cell lines, which means that healthy tissue could also be affected during efficient treatment of an in vivo tumor. This problem might be solved via surface modifications of liposomes with antibodies or antibody fragments to specifically target cancer cells [44]. For a more reliable evaluation of the cytotoxic efficiency of Cu(DDC)_2_ liposomes and heterotypic 3D-LS-hSkMC and 3D-LS-hDFa co-culture spheroids were used as cell models. The achieved EC_50_ values of LS co-culture spheroids did not differ from those of LS monoculture spheroids. Furthermore, the same morphological changes of LS monoculture spheroids after treatment with Cu(DDC)_2_ liposomes were observed for LS co-culture spheroids. It was expected, that hSkMC and/or hDFa eventually influence the spheroid resistance towards Cu(DDC)_2_ liposomes, which was not observed in this study. In particular fibroblasts, as cancer associated fibroblasts (CAFs), have tumor-promoting effects, supporting the proliferation and metastasis of cancer cells and enhancing resistances against anticancer drugs [45]. Attention should be paid to the fact, that fibroblasts are a highly heterogeneous cell population with phenotypic characteristics, which strongly differ from where they originate [46]. It could be possible, that the hDFa used in this study were not suitable to mimic CAFs in 3D spheroids. Therefore, a combination of fibroblasts and myoblasts is suggested to be ideal for the generation of CAFs [47].

Further work is considered to be of interest for the future. This includes the transfer of the liposome preparation process from lab scale to large scale. Different preparation techniques, such as high pressure homogenization [48], microfluidization [49], cross flow filtration [50], or dual asymmetric centrifugation [51,52] can be implemented for the production of large volumes of Cu^2+^ liposomes. Formation of Cu(DDC)_2_ complexes inside the liposomes and separation of non-encapsulated material may be achieved by using industrial high throughput columns and high pressure filtration under aseptic conditions. 

Additional neuroblastoma cell lines might be of interest for further development and formation of 3D spheroids. Labeling of liposomal components and fluorescence activated cell sorting after incubation and later destruction of the spheroid might be helpful to gain more knowledge about the distribution and intraspheroidal processing of the liposomes and its contents in the different regions of the spheroid.

## 5. Conclusions

This study shows that the treatment of 3D spheroids, compared to 2D monolayers, provides more information concerning the cytotoxic effectiveness of Cu(DDC)_2_ liposomes. Accordingly, it was proven that Cu(DDC)_2_ liposomes effectively reduced the viability of in vitro neuroblastoma 3D spheroids. As it turned out that primary surrounding cells will also be affected, the implementation of a targeting structure on the nanocarriers seems to be of importance for reducing unwanted side effects and will, therefore, be in the focus of further studies.

## Figures and Tables

**Figure 1 pharmaceutics-13-00894-f001:**
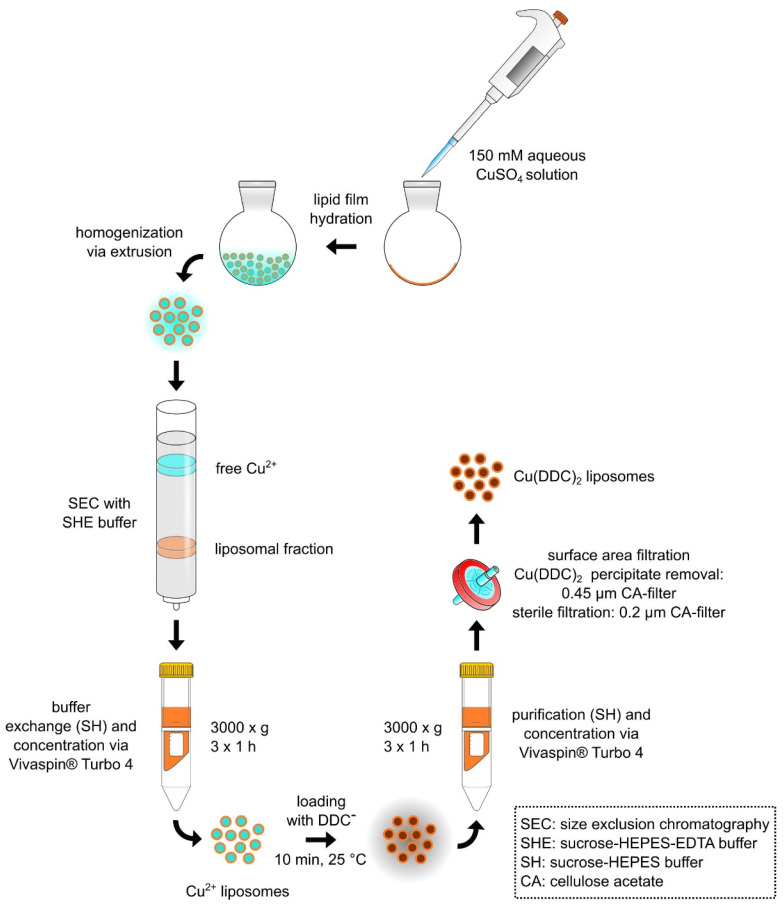
Schematic representation of Cu(DDC)_2_ liposome preparation by generating Cu^2+^ liposomes followed by remote loading of DDC^−^. This method was modified from Wehbe et al. (2016) [8].

**Figure 2 pharmaceutics-13-00894-f002:**
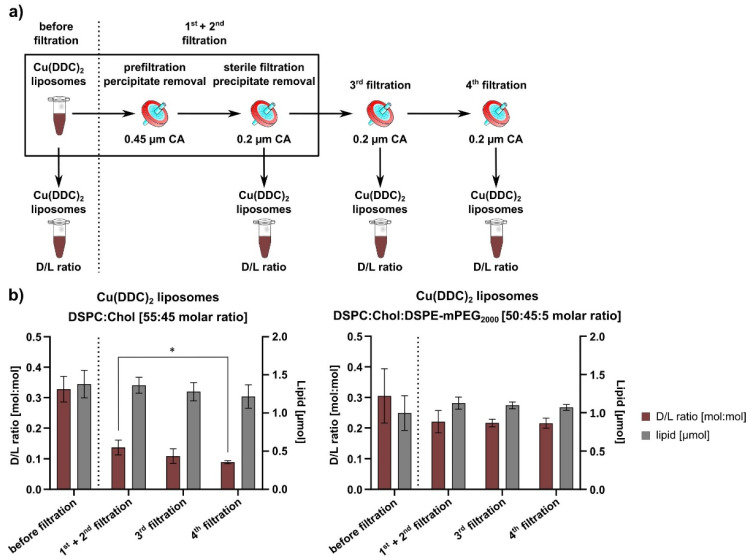
Validation analysis of the removal of extraliposomal Cu(DDC)_2_ via surface area filtration. (**a**) Experimental procedure of testing consecutive filtration steps for the removal of extraliposomal Cu(DDC)_2_. The 1^st^ + 2^nd^ filtration was part of the liposome preparation process. The 3^rd^ and 4^th^ filtration were conducted to determine the limit of filtration steps until affecting liposomal stability. Aliquots were collected after each step to determine the D/L ratio and lipid amount. (**b**) The D/L ratio and lipid amount of non-PEGylated and PEGylated liposomes were analyzed after consecutive filtration steps and evaluated statistically via one-way ANOVA and Tukey’s multiple comparison test (* *p* = 0.0104). Data are expressed as the mean ± SD (*n* = 3–4).

**Figure 3 pharmaceutics-13-00894-f003:**
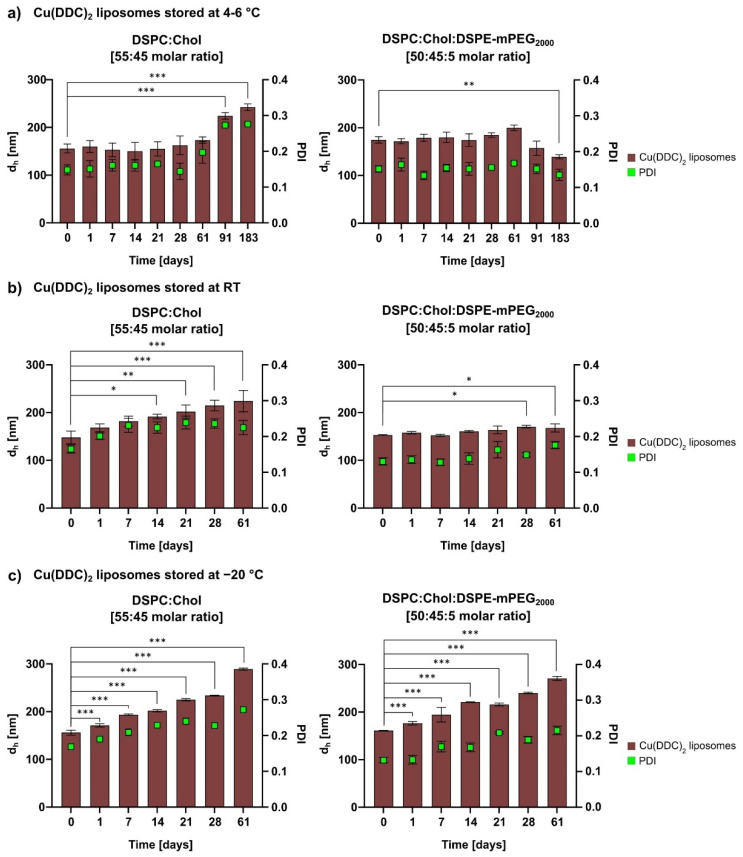
Colloidal stability analysis of Cu(DDC)_2_ liposomes after different storage conditions. Aliquots of non-PEGylated and PEGylated Cu(DDC)_2_ liposomes were analyzed via dynamic light scattering (DLS) over 6 months (183 days) at 4–6 °C (**a**) and over 2 months (61 days) at RT (**b**) and −20 °C (**c**). Significant differences were evaluated via one-way ANOVA and Dunnett’s post-hoc test (* *p* ≤ 0.05; ** *p* ≤ 0.01; *** *p* ≤ 0.001). Data are expressed as the mean ± SD (*n* = 3).

**Figure 4 pharmaceutics-13-00894-f004:**
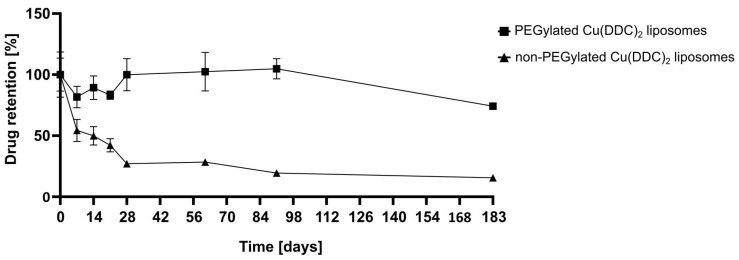
Liposomal Cu(DDC)_2_ retention upon storage at 4–6 °C. Aliquots of non-PEGylated and PEGylated Cu(DDC)_2_ liposomes (8 mM total lipid) were stored at 4–6 °C for 6 months (183 days). Samples were filtered through a 0.2-µm CA filter after specific storage time points. The D/L ratio of Cu(DDC)_2_ liposomes was determined before and after the filtration step. A reduction of the D/L ratio indicated Cu(DDC)_2_ release from liposomes. Data are expressed as the mean ± SD (*n* = 3).

**Figure 5 pharmaceutics-13-00894-f005:**
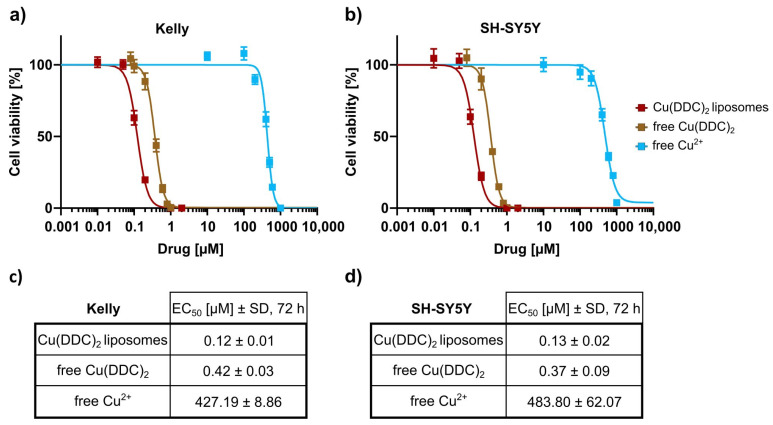
Determination of the cell viability reduction of Cu(DDC)_2_ liposomes and the free substances Cu(DDC)_2_ and Cu^2+^ on 2D monoculture neuroblastoma cell models. (**a**,**b**) 2D Kelly (**a**) and SH-SY5Y (**b**) monolayers were treated with liposomal Cu(DDC)_2_ (DSPC:Chol [55:45 molar ratio]), free Cu(DDC)_2_ and free Cu^2+^. Cell viability curves were obtained using 2D CellTiter-Glo^®^ assay after 72 h of treatment. Data are expressed as the mean ± SD (*n* = 3–4). (**c**,**d**) EC_50_ values after 72 h of treatment with liposomal Cu(DDC)_2_, free Cu(DDC)_2_, and free Cu^2+^. Data are expressed as the mean ± SD (*n* = 3–4).

**Figure 6 pharmaceutics-13-00894-f006:**
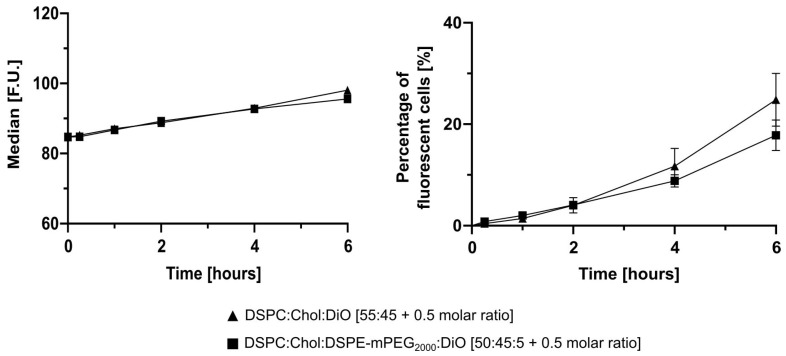
Cellular uptake of DiO labelled liposomes by 2D LS monocultures. 2D LS monolayers were incubated with non-PEGylated and PEGylated DiO labelled liposomes for 15 min, 1, 2, 4, and 6 h. Cellular uptake was evaluated via flow cytometry. The percentage of fluorescent cells (living gate) showed 24.8 ± 5.2% uptake for non-PEGylated and 17.8 ± 3.0% for PEGylated DiO liposomes. Data are expressed as the mean ± SD (*n* = 3).

**Figure 7 pharmaceutics-13-00894-f007:**
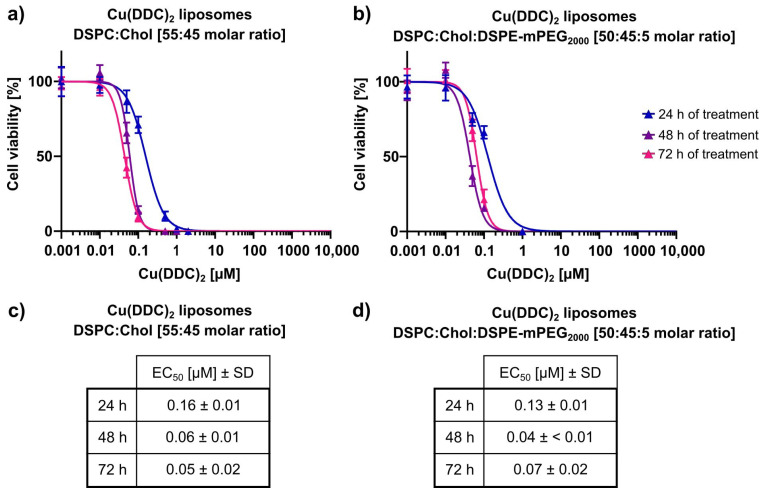
In vitro cytotoxicity of liposomal Cu(DDC)_2_ on 2D LS monoculture cells. (**a**,**b**) LS monolayers were incubated with non-PEGylated (**a**) and PEGylated (**b**) Cu(DDC)_2_ liposomes for 24, 48, and 72 h. Cell viability was analyzed using the 2D CellTiter-Glo^®^ assay after indicated treatment duration. (**c**,**d**) EC_50_ values after 24, 48, and 72 h of treatment with non-PEGylated (**c**) and PEGylated (**d**) Cu(DDC)_2_ liposomes. Data are expressed as the mean ± SD (*n* = 3–7).

**Figure 8 pharmaceutics-13-00894-f008:**
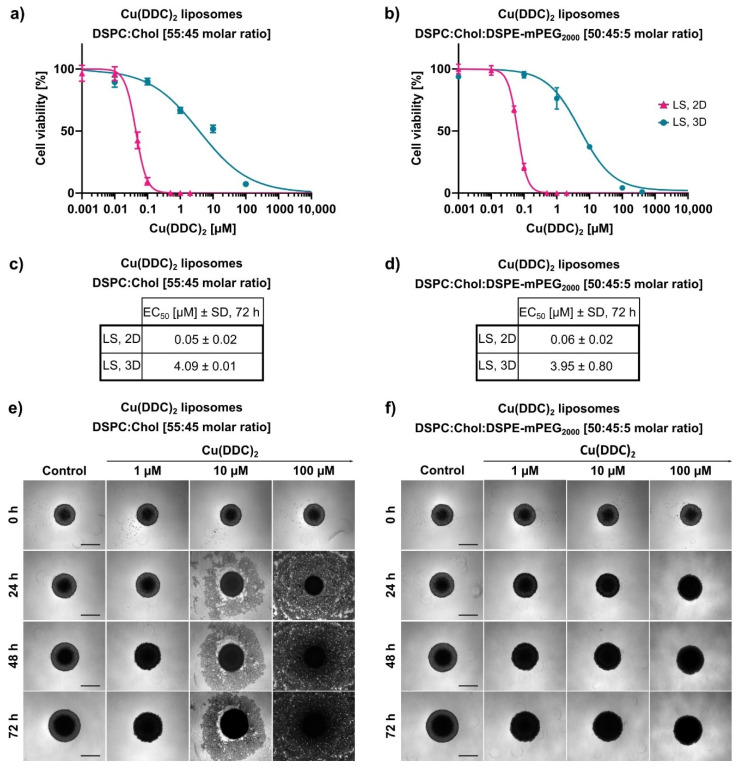
In vitro cytotoxicity of Cu(DDC)_2_ liposomes on neuroblastoma 2D and 3D cell cultures. (**a**,**b**) LS monolayers (2D) and LS monoculture spheroids (3D) were treated 72 h with non-PEGylated (**a**) and PEGylated (**b**) Cu(DDC)_2_ liposomes. 3D spheroids were generated by using agarose-coated wells with subsequent centrifugation. Cell viability curves were obtained using 2D and 3D CellTiter-Glo^®^ assay, respectively after indicated treatment duration. Data are expressed as the mean ± SD (LS, 2D: *n* = 7; LS, 3D: *n* = 5). (**c**,**d**) EC_50_ values after 72 h of treatment with non-PEGylated (**c**) and PEGylated (**d**) Cu(DDC)_2_ liposomes. Data are expressed as the mean ± SD (*n* = 5–7). (**e**,**f**) Representative brightfield microscopy images of LS monoculture spheroids. Images were taken after 24, 48, and 72 h of treatment with non-PEGylated (**e**) and PEGylated (**f**) Cu(DDC)_2_ liposomes, respectively. Scale bar = 400 µm.

**Figure 9 pharmaceutics-13-00894-f009:**
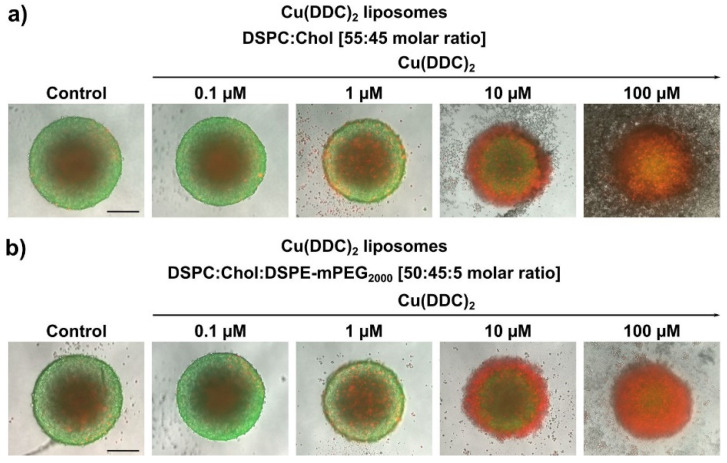
Representative fluorescence microscopy images of 3D LS monoculture spheroids on the basis of live/dead staining. 3D LS spheroids were treated for 72 h with non-PEGylated (**a**) and PEGylated (**b**) Cu(DDC)_2_ liposomes. The spheroids were stained with a combination of two dyes: 1 µM calcein-AM and 5 µM PI, added 2 h prior to the end of liposomal treatment. Calcein-AM (green) is visible at the outer cell layer, indicating highly proliferating and viable cells. PI (red) is primarily located in the inner core, indicating necrotic and dead cells. With increasing doses of Cu(DDC)_2_ liposomes, the PI signal intensified in the outer proliferating cell layer, until reaching the necrotic core. Scale bar = 200 µm.

**Figure 10 pharmaceutics-13-00894-f010:**
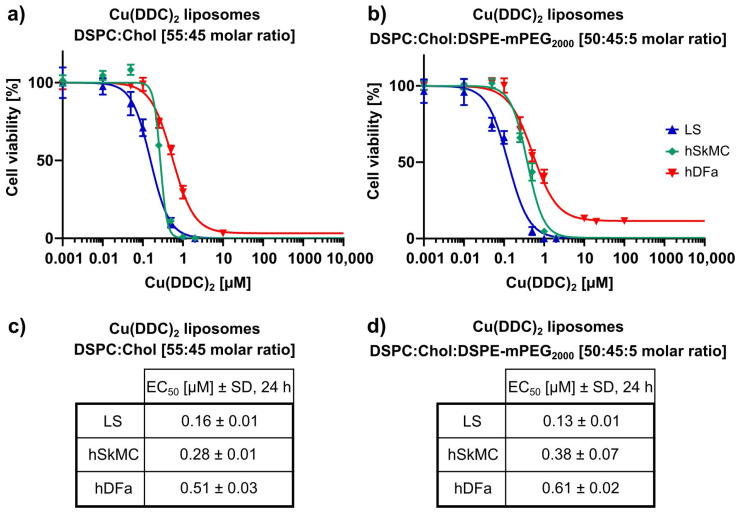
In vitro cytotoxicity of Cu(DDC)_2_ liposomes on 2D hSkMC and 2D hDFa monoculture cell models. (**a**,**b**) 2D monolayers of LS, hSkMC, and hDFa cells were incubated with non-PEGylated (**a**) and PEGylated (**b**) Cu(DDC)_2_ liposomes for 24 h. Cell viability was analyzed using the 2D CellTiter-Glo^®^ assay after indicated treatment duration. (**c**,**d**) EC_50_ values after 24 h of treatment with non-PEGylated (**c**) and PEGylated (**d**) Cu(DDC)_2_ liposomes. Data are expressed as the mean ± SD (*n* = 3).

**Figure 11 pharmaceutics-13-00894-f011:**
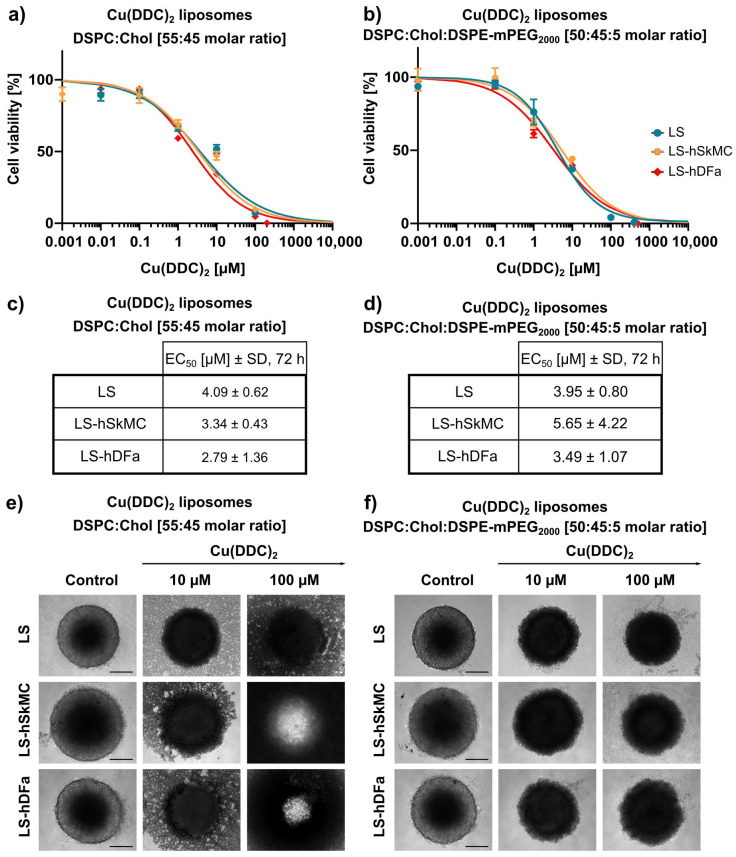
In vitro cytotoxicity of Cu(DDC)_2_ liposomes on LS-hSkMC and LS-hDFa co-culture spheroids. (**a**,**b**) LS monoculture spheroids, LS-hSkMC and LS-hDFa co-culture spheroids were treated 72 h with non-PEGylated (**a**) and PEGylated (**b**) Cu(DDC)_2_ liposomes. 3D spheroids were generated by using agarose-coated wells with subsequent centrifugation. hSkMC and hDFa cells are mixed with LS cells in a 1:1 ratio for co-culture spheroids, respectively. Cell viability curves were obtained using the 3D CellTiter-Glo^®^ assay after indicated treatment duration. (**c**,**d**) EC_50_ values after 72 h of treatment with non-PEGylated (**c**) and PEGylated (**d**) Cu(DDC)_2_ liposomes. Data are expressed as the mean ± SD (*n* = 3). (**e**,**f**) Representative brightfield microscopy images of LS monoculture and co-culture spheroids with hSkMC and hDFa, respectively. Images were taken after 72 h of treatment with non-PEGylated (**e**) and PEGylated (**f**) Cu(DDC)_2_ liposomes. Scale bar = 200 µm.

**Table 1 pharmaceutics-13-00894-t001:** Hydrodynamic diameter (d_h_), polydispersity index (PDI), and D/L ratio of prepared liposomes. Data are expressed as the mean ± SD (*n* = 7).

Liposomes	d_h_ [nm]	PDI	D/L Ratio [mol:mol]
Non-PEGylated Cu^2+^ liposomes (DSPC:Chol)	119 ± 3	0.06 ± 0.02	-
PEGylated Cu^2+^ liposomes (DSPC:Chol:DSPE-mPEG_2000_)	116 ± 5	0.08 ± 0.02	-
Non-PEGylated Cu(DDC)_2_ liposomes (DSPC:Chol)	156 ± 7	0.16 ± 0.02	0.15 ± 0.03
PEGylated Cu(DDC)_2_ liposomes (DSPC:Chol:DSPE-mPEG_2000_)	161 ± 7	0.14 ± 0.01	0.30 ± 0.04

## Data Availability

Data sharing not applicable.

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
