# Peer review of "Preclinical In Vitro Studies with 3D Spheroids to Evaluate Cu(DDC)2 Containing Liposomes for the Treatment of Neuroblastoma"

_pharmaceutics, 2021, doi:10.3390/pharmaceutics13060894_

Round 1

Reviewer 1 Report

Hartwig and colleagues explore the preclinical effects of Liposomal Cu(DDC)2 formulation in a 3D in vitro model of neuroblastoma. The overall study is well presented and the experiments conducted appropriately. However, this work is entirely based on a single neuroblastoma 3D spheroid culture model. For example, the SH-SY5Y neuroblastoma cell line has the capability to form spheroids but in the present study it has not been tested as 3D culture. To further conclude that the liposomes are effective on neuroblastoma 3D microtumors, the authors should confirm their efficacy in at least one different neuroblastoma cell line, or primary neuroblastoma cells with sphere-formation capability.

Some other minor considerations below:

-Line 78: is there a rationale behind the tumor type selection? Why did you choose LS neuroblastoma cell line to perform this study? The connection to the next paragraph seems leaky and can be improved/reorganized.

Line 213: can the authors specify the medium used for 3D cell co-cultures?

-Line 221. did the authors test the viability  of 3D LS NB spheroids only with Cu only?

-Line 242: the different concentrations used might be indicated also in the corresponding results (line 368).

-Line 368: did the authors measure the cellular uptake of DiO labeled liposomes in 3D LS spheroids?

-Line 415: change the term amazingly 

Line 427: explain what n=5-7 means

-Line 452: the authors define the outer layer of non PEGylated liposomes at higher concentrations as PI stained and necrotic. However, no stain is detectable in Figure 9a (fifth image) in cells detached from the spheroid. Can the authors provide higher resolution/magnification of the sphere details?

Line 477: the charachterization of the 3d co-culture can be improved:  after 72h of cultures, how neuroblastoma-hSkMC are organized? Is the initial 1:1 ratio maintained over culture? 

Line 480: do these cells activate apoptosis of Ros production in response to this treatment? 

Line 494: Why bright field images appear as in Figure 11e, 100uM treatment? Better images required.

Line 509: can the authors add a comment about the possibility of scaling-up this synthesis preparation process? 

Line 632: change the term microtumors into 3d in vitro spheroids or analogues.

Reviewer 2 Report

Reviewer’s Comments:

The manuscript "Preclinical in vitro studies with 3D spheroids to evaluate Cu(DDC)2 containing liposomes for the treatment of neuroblastoma" by Hartwig et al demonstrated that shows that the treatment of 3D spheroids, compared to 2D monolayers, provides more information concerning the cytotoxic effectiveness of Cu(DDC)2 liposomes.

Comments:

  1. Please include the catalog numbers of kits and reagents used in this study.
  2. Please explain asterisks (p-value range) in the figure legends.
  3. Please add a brief paragraph on “future directions to this study” at the end of the discussion/conclusions section.
  4. Please be consistent with the style of references.

Reviewer 3 Report

The research article entitled “Preclinical in vitro studies with 3D spheroids to evaluate Cu(DDC)2 containing liposomes for the treatment of neuroblastoma” by Friederike Hartwig, Monika Köll-Weber and Regine Süsscontains original findings, its structure contains all the key elements of obligatory sections such as Abstract, Introduction, Material and Methods, Results, Discussion and References as well.

The choice of topic is very topical and extremely interesting. The use of liposomes is very hot field- see eg new vaccine developments and mRNA technologies even in oncology. The Introduction is logical, very concise. However, liposomes and 3D neuroblastoma systems could be construed in a few sentences to make the importance of their own results more expressive.

The Materials and Methods section is accurate and traceable. Microscopic examinations could be given in more detail here, as the type of system may limit the actual results and measurement limits (see later). This must be revised.

The Results are well summarized, understandable, and the statistical analysis is adequate. However, it is not enough to focus only on cell viability with few methods, since in the case of 3D Live / dead staining, the necrotic cells do not necessarily mean dead cells. The exact location inside the spheroids is not known, it would be good to know if sections were made or a Z stack was made to get a more accurate picture. I would definitely suggest additional measurements to get more accurate results (WB or histochemistry). To compare 2D and 3D systems phenotype and metabolic activity must be compared as well.

The Discussion does not go far enough to compare the results of others on this needs to be improved. The discussion does not go far enough to compare the results of others on this needs to be improved. More relevant references are needed for this section.

Overall very good work, but needs to be improved for better quality for readers.

Round 2

Reviewer 1 Report

The manuscript by Hartwig and collegues provides a comprehensive charachterization of the effects of Cu(CCD)2 liposomes on a neuroblastoma cell line, showing its efficacy in inducing cell toxicity.

After reviewing the manuscript, I would recommend the publication in the present form.